# Dynamics and heterogeneity of a fate determinant during transition towards cell differentiation

Nicolás Peláez[1,2], Arnau Gavalda-Miralles[2†], Bao Wang[3], Heliodoro Tejedor Navarro[2], Herman Gudjonson[4], Ilaria Rebay[5], Aaron R Dinner[4], Aggelos K Katsaggelos[3], Luís AN Amaral[2,6*], Richard W Carthew[1*]

[1]Department of Molecular Biosciences, Northwestern University, Evanston, United States; [2]Department of Chemical and Biological Engineering, Howard Hughes Medical Institute, University Northwestern, Evanston, United States; [3]Department Electrical Engineering and Computer Science, Northwestern University, Evanston, United States; [4]James Franck Institute, University of Chicago, Chicago, United States; [5]Ben May Department for Cancer Research, University of Chicago, Chicago, United States; [6]Department of Physics and Astronomy, Northwestern University, Evanston, United States

**Abstract** Yan is an ETS-domain transcription factor responsible for maintaining Drosophila eye cells in a multipotent state. Yan is at the core of a regulatory network that determines the time and place in which cells transit from multipotency to one of several differentiated lineages. Using a fluorescent reporter for Yan expression, we observed a biphasic distribution of Yan in multipotent cells, with a rapid inductive phase and slow decay phase. Transitions to various differentiated states occurred over the course of this dynamic process, suggesting that Yan expression level does not strongly determine cell potential. Consistent with this conclusion, perturbing Yan expression by varying gene dosage had no effect on cell fate transitions. However, we observed that as cells transited to differentiation, Yan expression became highly heterogeneous and this heterogeneity was transient. Signals received via the EGF Receptor were necessary for the transience in Yan noise since genetic loss caused sustained noise. Since these signals are essential for eye cells to differentiate, we suggest that dynamic heterogeneity of Yan is a necessary element of the transition process, and cell states are stabilized through noise reduction.

**\*For correspondence:** amaral@ northwestern.edu (LAA); r-carthew@northwestern.edu (RWC)

**Present address:** [†]Department of Chemical Engineering, Universitat Rovira i Virgili, Tarragona, United States

**Competing interests:** The authors declare that no competing interests exist.

## Introduction

Cells within complex organisms exist in different states that confer specific functionalities to each cell. Cellular states can be organized into cyclic cascades such as the G1-S-G2-M cell cycle, or into linear cascades as observed in cell differentiation. A common feature of cells that undergo state transitions is the apparent irreversibility of the transitions even when such transitions are triggered by transient stimuli. Modeling of these transitions often assumes system bistability, in which cells can be resting in one of two stable states.

Animal cells frequently utilize transcription factors to enforce a given state, and transitions to another state are driven by increasing or decreasing the levels of these transcription factors (*Yao et al., 2008*; *Kueh et al., 2013*; *Laslo et al., 2006*; *Park et al., 2012*). The Rb-E2F pathway generates bistability in E2F expression, which dictates the transition from G1 to S phase (*Yao et al., 2008*). Expression of the transcription factor PU.1 determines lymphoid versus myeloid hematopoietic cell lineages (*Kueh et al., 2013*; *Laslo et al., 2006*). Adipocyte differentiation is controlled by

**eLife digest** As animal cells develop, they pass through different states to mature into specific cell types. For some cells, this development depends on the cell's ability to switch between two stable states, a property called bistability.

Many bistable systems operate during development and often feature proteins called transcription factors that regulate a few cell states in specific tissues. These proteins activate or repress a range of target genes, and cells can adjust the levels of the transcription factors to bring about transitions between states. In fruit flies, two transcription factors, called Yan and Pnt, regulate numerous processes throughout development.

In the developing embryo of a fruit fly, Yan and Pnt are regulated by signals induced by a receptor called EGFR. When EGFR is activated, Pnt is produced and Yan is degraded. When this receptor is not activated, Yan is produced and represses Pnt. Mathematical modelling of these interactions has concluded that this is a bistable system: that is, cells should either have high levels of Yan and low levels Pnt, or low Yan levels and high Pnt levels. However, larval eye cells first activate, and then turn off, both proteins together. This argues against bistability and raises questions about how these proteins regulate cell fate transitions in the eye, and perhaps in other organs.

To investigate this question, Peláez et al. tagged Yan with a fluorescent marker to track its activity in the eyes of fruit fly larvae as they develop. A combination of fluorescence-based microscopy and an automated imaging analysis system were then used to score fluorescence in thousands of individual eye cells and assess the changes in the levels of Yan over time. This approach revealed some unexpected results.

Yan levels were seen to vary in both immature and maturing cells. Thus eye cells transition between states as Yan levels increase rapidly, suggesting the need for a mechanism that is distinct from bistability. Peláez et al. suggest that larger changes in the seemingly random fluctuations in the levels of the transcription factor (also known as "expression noise") might play a role in this mechanism. In particular, Yan expression noise briefly peaks as cells transition to a more mature state, and the transient nature of this 'noisy' response requires the activation of EGFR.

One possible explanation for these observations is that Yan's effect on cell states depends on this variability in its levels, which might prime cells to change states when they receive another signal. These findings also raise many questions for future studies to explore, including how this increase in the noise level of Yan is triggered as cells begin their transitions towards specific cell types.

differential expression of C/EBP and PPARγ proteins (*Park et al., 2012*). Regulation by positive feedback is a hallmark of bistable systems, and in all of the above cases, the transcription factors act in one or more positive feedback circuits. In some systems, positive feedback is generated by two transcription factors that mutually repress each other's expression. In these scenarios, cells in one state continually express a transcription factor that represses a second transcription factor, and when these cells transit to another state, they continually express the second transcription factor, which represses its antagonist. Fate restriction in hematopoietic, neural, pancreatic, and muscle cell lineages is regulated by such double-negative feedback circuits (*Briscoe et al., 2000*; *Laslo et al., 2006*; *Olguin et al., 2007*; *Schaffer et al., 2010*).

Most examples of this type of bistable mechanism have transcription factors that act specifically within a handful of cell states limited to a single tissue or organ system. One remarkable exception to this rule is found in *Drosophila*. There, two ETS-domain transcription factors act in a wide assortment of cell types across the body and across the life cycle. Yan and Pointed (Pnt) act downstream of signals mediated by receptor tyrosine kinases (RTKs) that regulate cell differentiation, migration, and division in tissues ranging from ovarian follicular cells, dorsal and ventral neuroectoderm, embryonic mesoderm, the embryonic trachea, and the post-embryonic compound eye (*Dumstrei et al., 1998*; *Gabay et al., 1996*; *Halfon et al., 2000*; *Jurgens et al., 1984*; *Morimoto et al., 1996*; *O'Neill et al., 1994*; *Ohshiro et al., 2002*; *Yao et al., 2008*) It is thought that Yan and Pnt control such diverse cell states by acting in concert with tissue-specific transcription factors to regulate transcription of appropriate target genes. For instance, transcription of *even-skipped* occurs in

mesoderm only if Yan/Pnt act in combination with Tinman and Twist proteins (*Halfon et al., 2000*), whereas transcription of *prospero (pros)* in the eye only occurs if Yan/Pnt act in combination with Lozenge, Sine Oculis, and Glass proteins (*Hayashi et al., 2008*; *Xu et al., 2000*).

Several tissues show mutually exclusive expression of Yan and Pnt, suggestive of cross-repression (*Boisclair Lachance et al., 2014*). The best characterized is the embryonic ventral ectoderm, where ventral-most cells express Pnt and more lateral cells express Yan (*Gabay et al., 1996*). This pattern is established by secretion of a ligand for the EGF Receptor (EGFR) from the ventral midline (*Golembo et al., 1996*). EGFR activation in nearby ventral cells induces the Ras-MAPK pathway to express Pnt and degrade Yan (*Gabay et al., 1996*; *Melen et al., 2005*). Cells with insufficient EGFR activation express Yan, which represses Pnt. Mathematical modeling has described this as a bistable system in which cells are either in a High Yan/Low Pnt state or a Low Yan/High Pnt state (*Graham et al., 2010*; *Melen et al., 2005*). Transition from one state to the other is ultrasensitive to the strength of EGFR activation (*Melen et al., 2005*).

Paradoxically, other *Drosophila* tissues show co-expression of Yan and Pnt (*Boisclair Lachance et al., 2014*). The larval eye is one such tissue. Retinal progenitor cells initiate expression of both proteins, and when they transit to differentiated photoreceptor fates, these cells reduce expression of both proteins. In contrast, when retinal progenitor cells transit to differentiated cone cell fates, they maintain their expression of both proteins. These observations are at odds with long-standing genetic studies that support a standard bistable mechanism acting in the eye (*Lai and Rubin, 1992*; *O'Neill et al., 1994*; *Rebay and Rubin, 1995*). Thus, new approaches to studying these transitions in the eye are needed.

Here, we have adopted a systems-level approach to study Yan dynamics in the larval eye. A yellow fluorescent protein (YFP) — based isoform of Yan was developed as a reporter for Yan protein levels. Fluorescence-based microscopic imaging of cells was coupled with automated high-throughput image analysis to score fluorescence in each cell and annotate the data in a quantitative and unbiased fashion. Yan exhibits monostability, both in progenitor and differentiating cells, with Yan levels varying in cells in either state. Cell state transitions occur independent of absolute Yan concentrations, suggesting that some other mechanism allows Yan to regulate transitions. One such mechanism might be the noise in Yan levels, which undergoes a transient spike as cells begin to transition to differentiated states. Loss of EGFR signaling, which prevents cells from differentiating, causes these cells to have prolonged noisy Yan expression, and suggests that Yan noise is key for cell state transitions in the eye.

## Results

The compound eye epithelium is established during embryogenesis as an internal disc of cells called the eye imaginal disc (*Wolff and Ready, 1993*). During the larval phase of the life cycle, the disc grows in size by asynchronous cell division. During the final 50 hr of the larval phase, a morphogenetic furrow (MF) moves across the eye disc from posterior to anterior (*Figure 1A,B*). All cells arrest in G1 phase within five cell diameters anterior to the furrow, and then as the furrow passes through them, periodic clusters of cells express the proneural gene *atonal* (*Jarman et al., 1994*). *Atonal* expression is subsequently restricted to one cell per cluster, which becomes the R8 photoreceptor. Each R8 cell then secretes an EGFR ligand that activates the receptor in neighboring cells and causes them to transit from multipotent progenitor to differentiated states (*Figure 1C*) (*Freeman, 1996*). Transitions occur in a sequence of symmetric pairs of multipotent progenitor cells that differentiate into R2/R5, R3/R4, and R1/R6 photoreceptors (*Figure 1C*) (*Wolff and Ready, 1993*). Thereafter, a single progenitor transits to a R7 photoreceptor fate followed by two pairs of cells, C1/C2 and C3/C4, that differentiate into cone cells. These cone cells are non-neuronal and form the simple lens that overlies each cluster of eight photoreceptors. The furrow induces the nearly simultaneous differentiation of a column of R8 cells, with repeated column inductions producing approximately 800 units or ommatidia as the furrow moves across the eye.

A central tenet of the bistable model of cell differentiation in the eye posits that differentiation is marked by a transition from high Yan protein levels in multipotent progenitor cells to low Yan levels in differentiating cells (*Graham et al., 2010*). Formulation of this model stemmed from studies of R7 cell differentiation, the final photoreceptor recruited to each ommatidium. Reduced Yan causes inappropriate expression of the R7 determinant *pros* and ectopic R7 cells in *yan* hypomorphic mutants

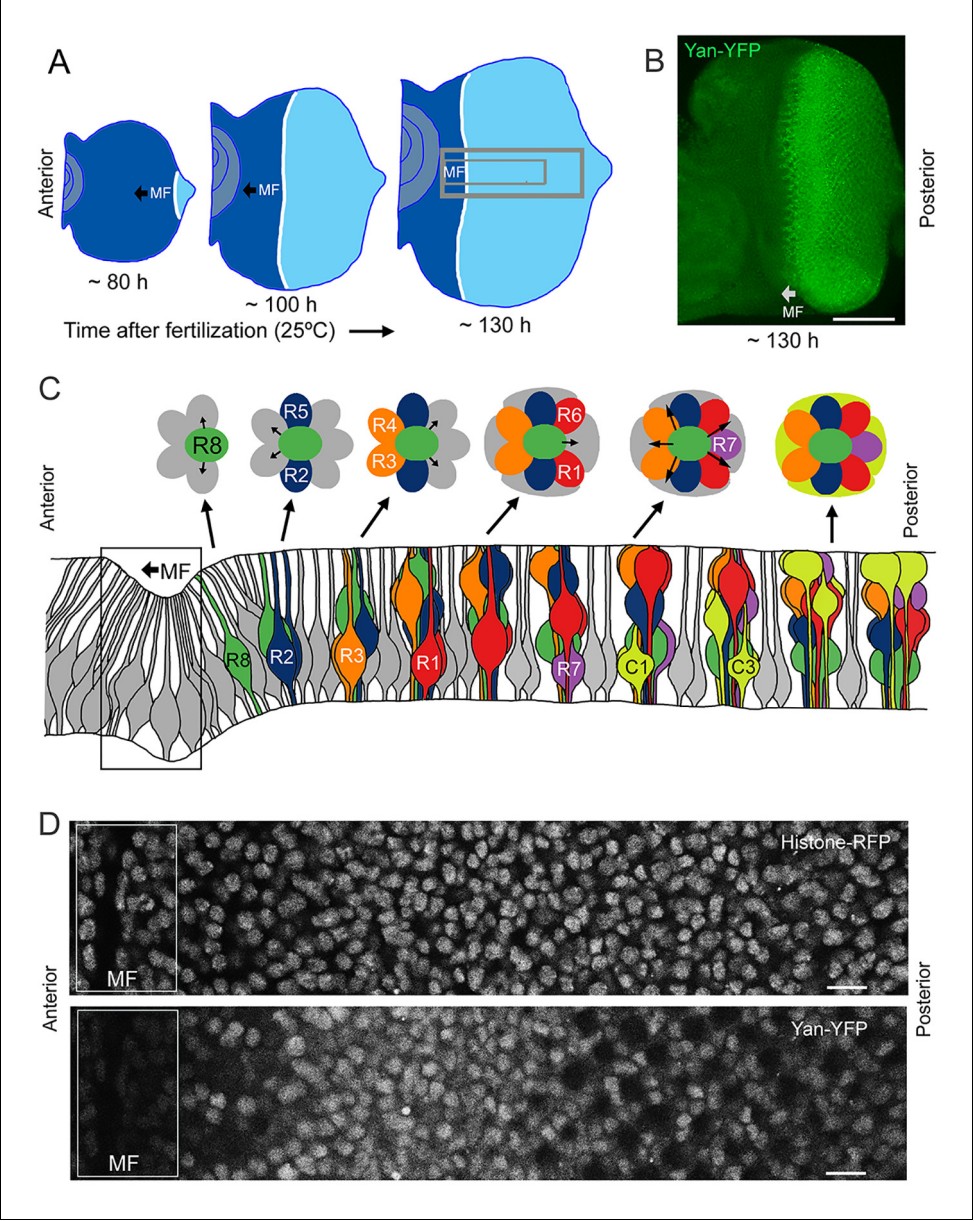

**Figure 1.** Development and patterning of the compound eye. (**A**) Differentiation is initiated in the developing eye by the MF, which moves across the eye epithelium. On the furrow's posterior side, G1-arrested progenitor cells undergo differentiation (light blue). On the anterior side, progenitor cells are still proliferating (dark blue). The large grey rectangle outlines the region that was analyzed for Yan levels; the small rectangle corresponds to the developmental sequence outlined in panel C (**B**) A maximal projection of Yan-YFP fluorescence in an eye. Bar = 100 μm. (**C**) Top, an apical view of the sequential differentiation of eight photoreceptor (R1-R8) and four cone cell types (C1-C4) from multipotent progenitor cells (grey) in an ommatidium. Arrows denote inductive signal transmitted from the R8 to activate EGFR on nearby cells. Bottom, a cross-section view through an eye disc adapted after *Wolff and Ready (1993)*. Note the progenitor cell nuclei are basally positioned, and as they transition into a differentiated state, their nuclei migrate apically. C1/C2 cells are positioned anterior and posterior in the ommatidium while C3/C4 cells are positioned equatorial and polar in the ommatidium. (**D**) Top, an optical slice of H2Av-mRFP fluorescence in an eye disc at a plane that bisects progenitor cell nuclei. Bottom, the same optical slice imaged for Yan-YFP fluorescence. Bars = 8 μm.

The following figure supplements are available for figure 1:

**Figure supplement 1.** Expression and activity of the Yan-YFP transgene.

Figure 1 continued

**Figure supplement 2.** Segmentation and identification of cell nuclei in eye discs.
**Figure supplement 3.** Accuracy of cell-type identification.
**Figure supplement 4.** His2Av-mRFP fluorescence properties.

(*Kauffmann et al., 1996*; *Lai and Rubin, 1992*). Conversely, a Yan isoform that is resistant to MAPK-dependent degradation, blocks R7 differentiation and *pros* expression (*Kauffmann et al., 1996*; *Rebay and Rubin, 1995*).

## Quantifying Yan dynamics

To quantitatively test the bistable model, we used BAC recombineering to insert fast-fold yellow fluorescent protein (YFP) in-frame at the carboxy-terminus of the *yan* coding sequence (*Webber et al., 2013*). The *Yan-YFP* transgene fully complemented null mutations in the endogenous *yan* gene and completely restored normal eye development (*Figure 1—figure supplement 1*), demonstrating that the YFP tag does not compromise Yan function and that all essential genomic regulatory sequences are included. The pattern of Yan-YFP protein expression was qualitatively similar to that of endogenous Yan (*Figure 1—figure supplement 1*).

We used histone His2Av-mRFP fluorescence in fixed specimens to mark all eye cell nuclei for automated segmentation (*Figure 1D—figure supplement 2*). Nuclei were manually classified into the different cell types of the eye, which is possible because every cell undergoing differentiation can be unambiguously identified by its position and nuclear morphology without the need of cell-specific markers (*Ready et al., 1976*; *Tomlinson, 1985*; *Tomlinson and Ready, 1987*; *Wolff and Ready, 1993*). To validate our identification of all cell types using this method, we cross-checked with specific cell-specific markers, and found that our classification was accurate over 98% of the time (*Figure 1—figure supplement 3*). Cells were scored for nuclear Yan-YFP concentration and their exact position within 3D coordinate space of each fixed eye sample (*Figure 1—figure supplement 2*). Yan-YFP protein levels were normalized to His2Av-mRFP, which provided some control over measurement variation (*Figure 1—figure supplement 4*). We then mapped each cell's spatial position in the x-y plane of the eye disc onto time. Two reasons made this possible. First, the furrow moves at an approximately constant velocity forming one column of R8 cells every two hours (*Basler and Hafen, 1989*; *Campos-Ortega and Hofbauer, 1977*). Second, each column of R8 cells induces the other cell fates at a constant rate (*Lebovitz and Ready, 1986*). Thus even in a fixed specimen, the temporal dynamics of cell state transitions are visible in the repetitive physical organization of ommatidia relative to the furrow (*Figure 1C*). Together these features allow the estimation of time based on a cell's position relative to the furrow (*Figure 2—figure supplement 1*). This can be applied repeatedly to hundreds of cells in a single sample, creating a composite picture of the dynamics (*Figure 2B–F*). Although the developmental progression of an individual cell cannot be measured by this approach, it nevertheless provides a dynamic view of hundreds of cells across a developing epithelium as they respond to signaling processes. From this information, individual cell behaviors can be reconstructed and modeled.

Yan-YFP expression in multipotent progentior cells was biphasic (*Figure 2B*). Cells anterior to the furrow expressed a basal level of Yan-YFP, but this level dramatically increased in cells immediately anterior to the furrow, starting eight hours before the first R8 cells were identifiable. Approximately eight hours after R8 definition, Yan-YFP levels peaked, and thereafter gradually decayed until Yan-YFP approached its basal level again. The results are surprising in two ways. First, Yan-YFP is not maintained at a stable steady state within progenitor cells, which would have been predicted by the bistable model. Rather, its dynamics are reminiscent of monostable responses to transient stimuli, with a single basal steady state. Second, at the peak of Yan-YFP expression, there is remarkably large heterogeneity in Yan-YFP levels across cells.

We also followed Yan-YFP dynamics in cells as they transited into a differentiated state and thereafter. Again, the results did not follow the expectations predicted by the bistable model. First, progenitors transited to a differentiated state at levels of Yan-YFP that varied, depending upon the type

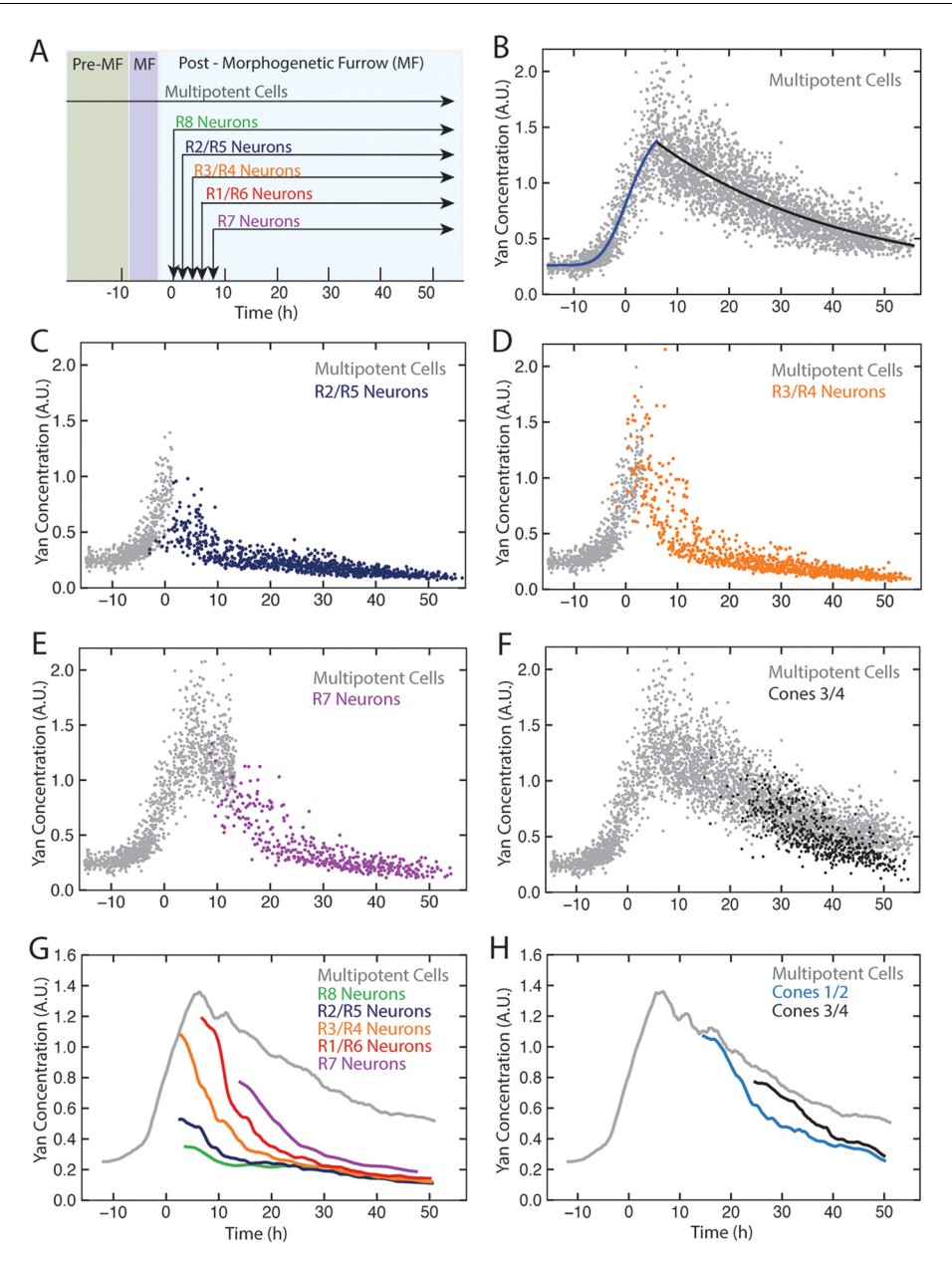

**Figure 2.** Dynamics of Yan-YFP in eye cells. (**A**) Average time at which initiation of differentiation is first detected by apical migration of committing cell nuclei. Time zero is set to when R8 differentiation initiates. Differentiation proceeds over a time course after commitment is initiated (horizontal arrows) (**B**) Yan-YFP fluorescence in multipotent progenitor cells. We fit a Hill function (blue curve) to the inductive phase and an exponential decay (black curve) to the decay phase. (**C-F**) Scatter plots of Yan-YFP levels in individual cells for R2/R5 (**C**), R3/R4 (**D**), R7 (**E**), and C3/C4 (**F**) cells. These are overlaid with scatter plots of Yan-YFP in multipotent cells at times preceding and coincident with the appearance of the relevant differentiated cells. Note the similar Yan-YFP levels between multipotent cells and differentiating cells at their first appearance. (**G**) Moving averages of Yan-YFP levels for multipotent and photoreceptor cells. Gaps between the multipotent and photoreceptor curves are due to the window size for line averaging. (**H**) Moving averages of Yan-YFP in multipotent and cone cells.

The following figure supplement is available for figure 2:

**Figure supplement 1.** Mapping identified nuclei within XY space of eye discs.

of differentiated state being adopted (*Figure 2C–H*). Cells entering the R3/R4 and R1/R6 states began with Yan-YFP levels that were approximately two-fold greater than cells entering the R2/R5 states. Cells entering the R7 state were intermediate between these two extremes. Despite these differences at the transition point, Yan-YFP levels decayed to a similar basal steady state irrespective of the photoreceptor type, and this basal level was at least three-fold lower than that which the progenitor cells achieved (*Figure 2G*). Thus, rather than the single high Yan progenitor state previously modeled, our results suggest a dynamic range of high Yan states from which different cell fates are specified according to the spatio-temporal sequence of differentiation.

We noted that for most photoreceptors, it took approximately 20 hr for Yan-YFP to decay to the basal steady state (*Figure 2G*), significantly longer than had been previously modeled (*Graham et al., 2010*). Since expression of several neural-specific genes is detected 2–8 hr after the transition (*Tomlinson and Ready, 1987*; *Van Vactor et al., 1988*), the slower than anticipated Yan decay indicates that early differentiation does not require cells to have assumed a basal steady-state level of Yan-YFP.

The last group of progenitors to differentiate form the non-neuronal cone cells. We also measured Yan-YFP in those cells. Yan-YFP dynamics in cone cells were more similar to progenitor cells over the same time period (*Figure 2F,H*). This behavior was in contrast to photoreceptors, which exhibited different decay dynamics from progenitor cells. Thus, accelerated degradation of Yan-YFP is not essential for cells to transition to all retinal cell states.

## EGFR-ras signaling regulates Yan-YFP dynamics

The bistable model posits that the switch from one state to another is triggered by a signal that progenitor cells receive though the EGFR protein. Given the unanticipated Yan-YFP dynamics, we wanted to ask whether and how they were influenced by EGFR signaling. *EGFR* null mutants are inviable; however, a temperature sensitive (ts) allele of *EGFR* enables controlled inactivation of the RTK's activity (*Kumar et al., 1998*). We grew *EGFR*[ts] mutant larvae at a restrictive temperature for 18 hr before analyzing effects on Yan-YFP. Surprisingly, progenitor cells exhibited biphasic expression of Yan-YFP over time, but the amplitude of the pulse in expression was significantly reduced (*Figure 3A*). This suggests that EGFR signaling contributes to the stimuli that induce the Yan-YFP peak. To further test this hypothesis, we misexpressed a constitutively active form of Ras protein in eye cells. The peak of Yan-YFP in progenitors was now prolonged (*Figure 3B*). Together, these results suggest that EGFR-Ras signaling stimulates the transient appearance of Yan-YFP in progenitor cells, and that the decline in Yan-YFP within older progenitor cells is linked to a loss of signal reception by these cells over time.

We next examined the effects of EGFR and Ras on Yan-YFP dynamics in cells as they differentiate. The bistable model predicts that EGFR is required for the loss of Yan-YFP in photoreceptors. Indeed, *EGFR*[ts] mutant R2/R5 cells delayed their initial decline in Yan-YFP levels (*Figure 3C*). Conversely, misexpression of constitutively active Ras caused a premature decline in Yan-YFP (*Figure 3D*). These results are consistent with EGFR-Ras stimulating the degradation of Yan-YFP as cells switch their states. However, Yan-YFP dropped to below-normal levels in *EGFR*[ts] mutant R2/R5 cells (*Figure 3C*). These complex effects suggest a dual role for EGFR signaling in photoreceptors. In the first few hours as cells transit to a photoreceptor state, EGFR stimulates the accelerated decay of Yan-YFP. Thereafter, EGFR inhibits the decay of Yan-YFP in a manner that might be related to that role that EGFR plays in boosting Yan-YFP levels in progenitor cells.

The source of EGFR ligand originates from the R8 cell (*Freeman, 1996*). If this diffusive ligand is responsible for controlling Yan-YFP levels in other photoreceptors as they are recruited to an ommatidium, we would predict a correlation between Yan-YFP levels in differentiating cells and their distances from adjacent R8 cells. Indeed, at the times when R2/R5 cells differentiate (~0–15 hr), their Yan-YFP levels are positively correlated (p<0.01) with their physical distance to the nearest R8 cells (*Figure 3—figure supplement 1*). These correlations are absent in *EGFR*[ts] mutants, providing evidence that R8 cells act through EGFR to control Yan-YFP dynamics in differentiating cells.

## Pnt regulates Yan dynamics

Pnt proteins have been hypothesized to cross-repress Yan expression, and this double negative feedback loop is thought to be necessary for bistability (*Graham et al., 2010*; *Shilo, 2014*). At odds

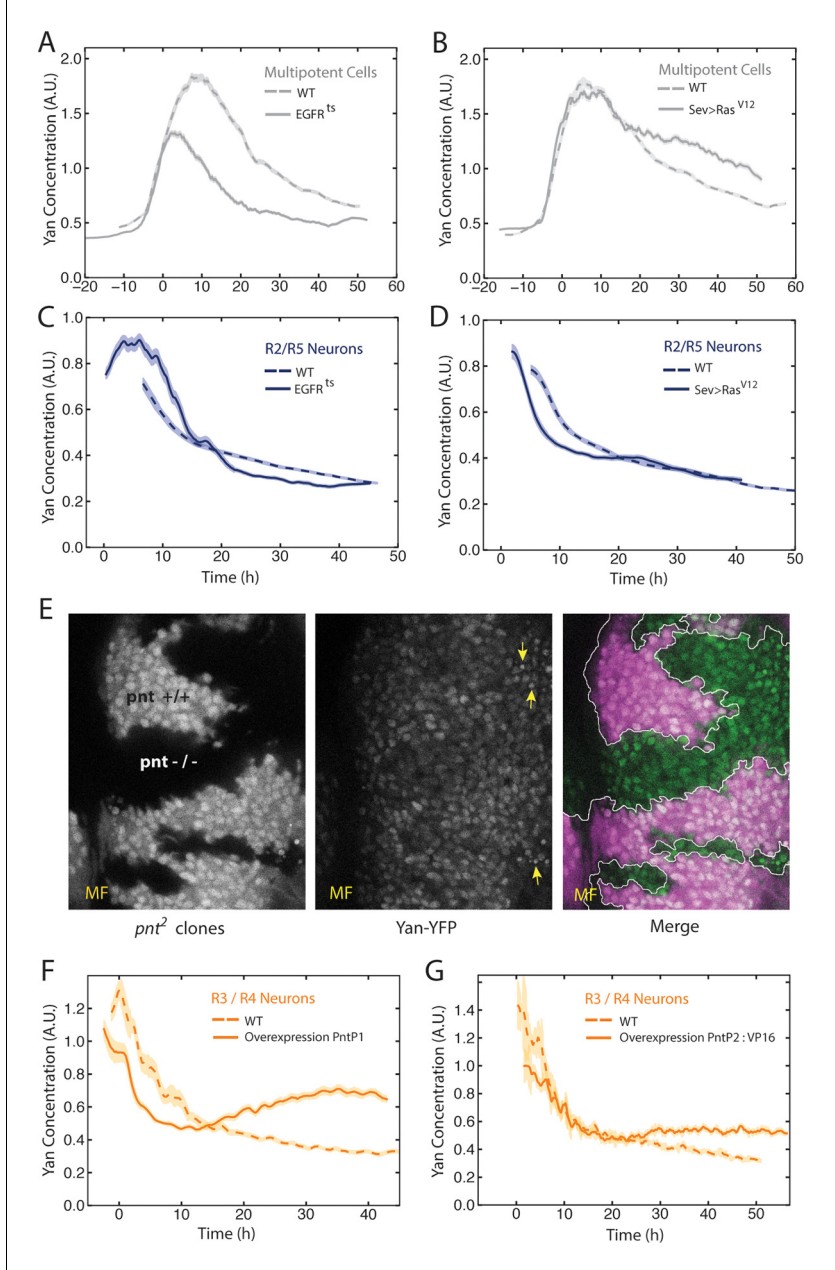

**Figure 3.** EGFR/Ras and Pnt regulate Yan-YFP levels. (A–D) Moving averages of Yan-YFP in different cell types. Shown with shading is the standard error of the mean for each moving average. (A,C) Wildtype and *EGFR^ts* mutants incubated at the non-permissive temperature and analyzed for progenitors (A) and R2/R5 cells (C). (B,D) Wildtype and *sev>Ras^{v12}* mutants were analyzed for progenitors (B) and R2/R5 cells (D). (E) Optical slice through progenitor cell nuclei in a disc that contains clones of *pnt^2* mutant cells. Left, fluorescence of RFP, which positively marks wildtype cells and not *pnt^2* mutant cells. Middle, Yan-YFP fluorescence, showing reduced levels in *pnt^2* mutant clones. Arrows highlight apoptotic nuclei. Right, merged image with Yan-YFP in green and RFP in purple. Clone borders are outlined. (F,G) Moving averages of Yan-YFP in R3/R4 cells that ectopically express PntP1 (F) or PntP2-VP16 (G) due to *LongGMR-Gal4* driving the UAS transgenes. Since PntP2 requires MAPK phosphorylation to become transcriptionally active, we misexpressed a VP16 fusion of PntP2. PntP1 is constitutively active (*Brunner et al., 1994*; *Gabay et al., 1996*).

The following figure supplement is available for figure 3:

**Figure supplement 1.** Yan-YFP levels in R2/R5 correlate with distance from R8 cells when EGFR is active.

with this view, Pnt and Yan proteins are co-expressed in progenitor and differentiating cells of the eye (*Boisclair Lachance et al., 2014*). Since Pnt proteins act downstream of many RTKs including EGFR, we wondered if Pnt mediated the positive effects of EGFR-Ras signaling on Yan-YFP in progenitor cells. Null mutants of the *pnt* gene are embryonic inviable; therefore we generated clones of eye cells that were null mutant for *pnt* in an otherwise wildtype animal. Mutant progenitor cells showed a reduction in Yan-YFP levels as they progressed through their biphasic expression pattern (*Figure 3E*). Thus, Pnt possibly mediates the positive effect of EGFR signaling on Yan-YFP expression in progenitors. We also wished to know if Pnt mediates the complex effects of EGFR in differentiating photoreceptors. Pnt proteins are rapidly cleared in photoreceptors (*Boisclair Lachance et al., 2014*) and so loss-of-function mutant analysis would be uninformative. Therefore, we overexpressed PntP1 or constitutively-active PntP2 in cells as they transited into a photoreceptor state and beyond. The early phase of Yan-YFP decay was accelerated while the later phase of Yan-YFP decay was inhibited (*Figure 3F,G*). These complex effects are precisely the opposite to those caused by loss of EGFR signaling, as would be expected if Pnt mediated EGFR's complex effects on Yan-YFP dynamics in photoreceptor cells.

## Cells are indifferent to absolute levels of Yan

The bistable model predicts that different cell states depend upon discrete absolute concentration of Yan present inside cells. To test this idea, we varied the number of *Yan-YFP* gene copies. In general, protein output is proportional to gene copy number in *Drosophila* (*Lucchesi and Rawls, 1973*). We increased *Yan-YFP* copy number from its normal diploid number to tetraploid, and monitored Yan-YFP in progenitors and differentiating cells. As expected, four copies caused a higher steady-state level of Yan-YFP in progenitor cells, though this increase was less than two-fold (*Figure 4A*). The amplitude of the Yan-YFP pulse was also increased as progenitor cells aged. Strikingly, as four-copy progenitor cells transited to a differentiated state, the onset of Yan-YFP decay occurred at the same time as it occurred for two-copy cells (*Figure 4A–C*). Yan-YFP levels were much greater in four-copy cells compared to two-copy cells making their transit into the identical cell states. To confirm that absolute Yan-YFP concentration had little effect on cell state transitions, we examined expression of a direct target of Yan in R7 cells: the *pros* gene (*Kauffmann et al., 1996*; *Xu et al., 2000*). Expression was monitored with an antibody specific for Pros protein. We observed at most a one hour delay in the onset of Pros expression in R7 cells containing four copies of *Yan-YFP* (*Figure 4D*), far less than the ten-hour delay predicted if absolute concentration of Yan-YFP dictated when Pros expression begins (*Figure 4D*).

Possibly, absolute concentration of Yan is unimportant when a cell transits to a different state, but the switch to a constant basal Yan level is robustly maintained regardless of starting concentration. An examination of Yan-YFP decay in photoreceptor cells makes that possibility less likely; Yan-YFP decays to different basal levels in two- versus four-copy differentiated cells (*Figure 4A–C*). To further test this notion, we fit the data to several plausible functional forms. We found that exponentially decaying functions systematically best-fit to the data (*Figure 4—figure supplement 1*). Thus, for each cell state we fit an exponential decay function to its Yan-YFP temporal profile (*Figure 4—figure supplement 2*). From these fits, we derived the average decay rate constants and half-lives of Yan-YFP for cells carrying two, four, or six copies of *yan*. As expected, we found that Yan-YFP half-life was different between progenitors and differentiating photoreceptors (*Figure 5—figure supplement 1*). The half-life in photoreceptors was two-fold lower than in progenitors, accounting for the more rapid loss of Yan-YFP in the former cells. Strikingly, Yan-YFP half-life was not significantly affected by *yan* copy number within either progenitor or photoreceptor cells (*Figure 5*). Thus, Yan-YFP concentration only affected its decay rate as a first-order function, implying that there is no higher order mechanism to accelerate Yan-YFP decay when cells contain greater concentrations of Yan-YFP.

As a final test of the effects of Yan-YFP levels on cell states, we allowed 4X and 6X *yan* animals to complete eye development and then examined the external patterning of the fully differentiated compound eye. The compound eyes were completely normal in appearance (*Figure 5—figure supplement 2*), implying that differentiation was unaffected by the absolute concentrations of Yan inside eye cells.

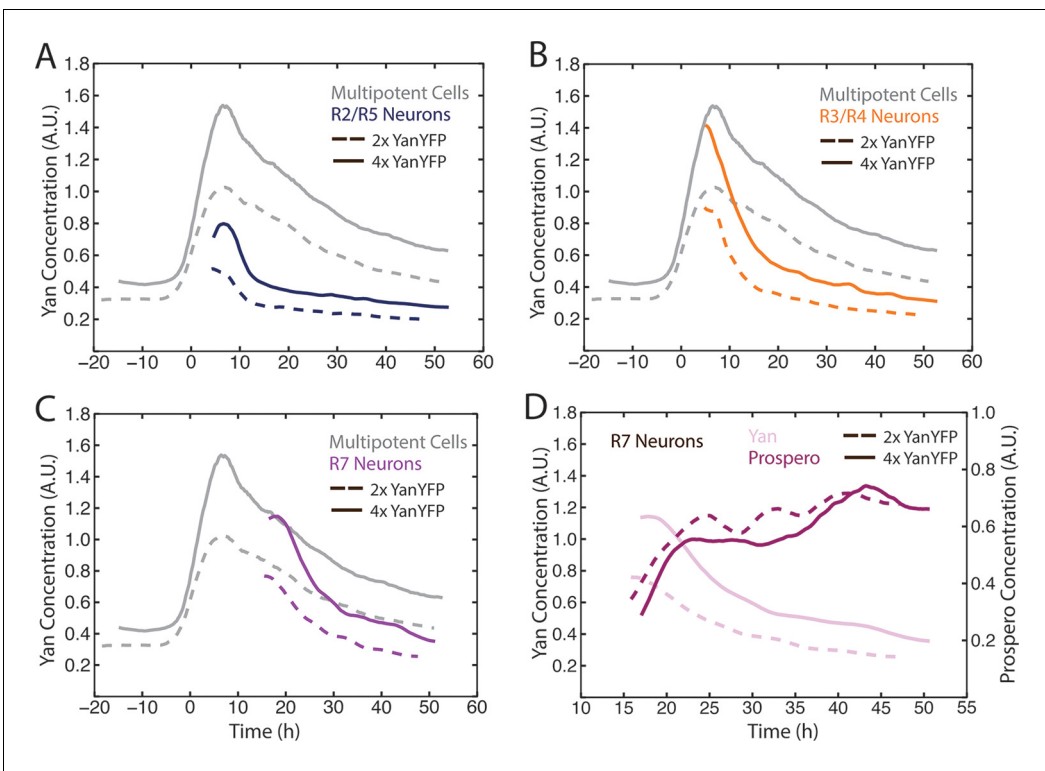

**Figure 4.** Cell state transitions are unaffected by Yan-YFP gene copy number. (A–C) Moving averages of Yan-YFP in eye discs containing two versus four copies of the *Yan-YFP* transgene. (A) R2/R5 and progenitor cells. (B) R3/R4 and progenitor cells. (C) R7 and progenitor cells. (D) Moving averages of Yan-YFP and Pros proteins in R7 cells containing either two vs. four copies of the *Yan-YFP* transgene.

The following figure supplements are available for figure 4:

**Figure supplement 1.** Model fitting Yan-YFP decay in eye disc cells.

**Figure supplement 2.** Exponential decay models.

## Yan expression noise spikes during cell state transitions

Our results indicate that Yan's effects on retinal cell states are not dictated by uniform and stable concentrations of Yan protein. One potential explanation is that Yan's effect on cell states actually depends on variability in Yan protein levels. A growing body of evidence is pointing to the importance of transient fluctuations in expression of factors to control cell states (*Cahan and Daley, 2013*; *Torres-Padilla and Chambers, 2014*). Key regulators of stem cells fluctuate in abundance over time, and this is correlated with stem cells existing in multiple connected microstates, with the overall structure of the population remaining in a stable pluripotent macrostate (*MacArthur and Lemischka, 2013*). Heterogeneity among cells is not simply due to independent noise in expression of individual genes but is due to fluctuating networks of regulatory genes (*Chang et al., 2008*; *Kumar et al., 2014*). Such fluctuations appear to be stochastic in nature, priming cells to transit into differentiated states with a certain probability that is guided by extrinsic signals.

Our data revealed considerable heterogeneity in the level of Yan-YFP among cells at similar developmental times (*Figure 2B*). To quantify the noise, we used developmental time to bin cells of similar age, and measured detrended fluctuations of Yan-YFP by calculating residuals to the fitted function and normalizing binned residuals to the function (*Goldberger et al., 2002*). Progenitor cells showed a spike in Yan-YFP noise as they began to induce Yan-YFP expression (*Figure 6A*). The noise spike was short-lived (approximately 17 hr), and noise thereafter returned to a basal level with

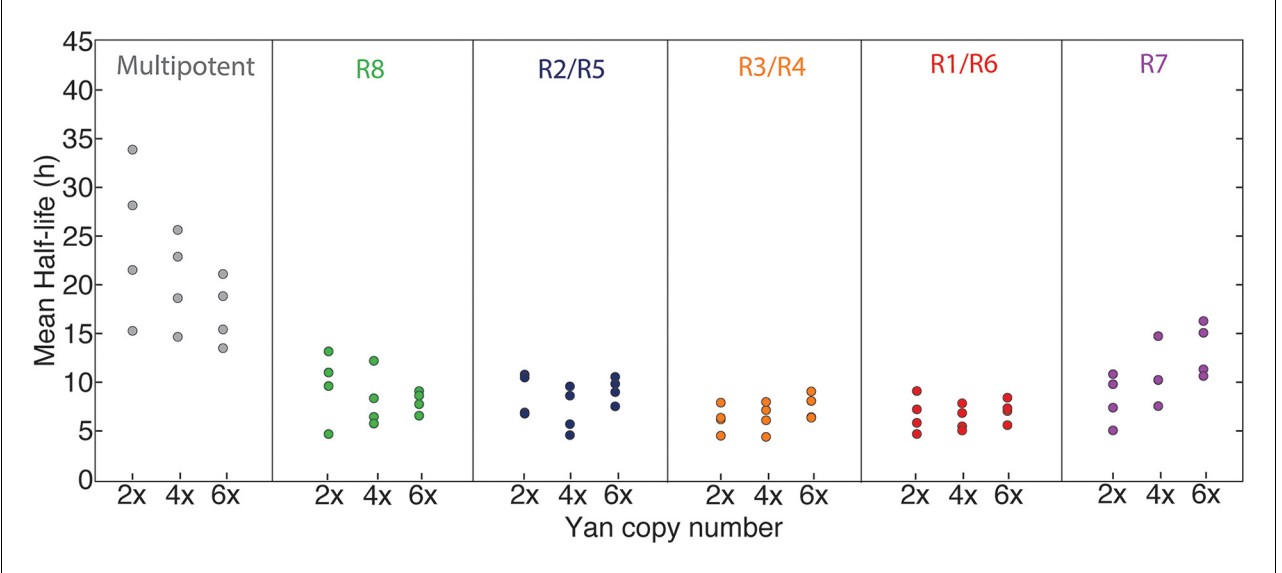

**Figure 5.** Yan protein half-life is largely unaffected by *yan* gene copy number. Exponential decay of Yan-YFP levels. Note the robustness of Yan-YFP half-lives across replicates and *yan* gene copy number. Note also how half-lives are nearly twice as long for progenitor cells versus photoreceptor neurons.

The following figure supplements are available for figure 5:

**Figure supplement 1.** Mean half-lives for Yan-YFP decay versus *yan* gene copy number (2, 4, 6x).

**Figure supplement 2.** Compound eyes of adults carrying two, four, or six copies of the yan gene.

secondary minor spikes. The major peak in noise magnitude coincided with the time at which R8 cells are formed.

Photoreceptor cells showed a large spike in Yan-YFP noise as they began to differentiate (*Figure 6B–E*). The magnitude of each noise peak varied with the photoreceptor cell state; R3 and R4 cells exhibited the greatest amplitude in noise (*Figure 6F*). These noise spikes showed a distinct temporal relationship, with spikes coinciding with the times at which individual cell states were switched (*Figure 6F*). Thus, the noise spikes are not a simple consequence of a global stimulus synchronously affecting noise in all cells. Thereafter, all cells reduced Yan-YFP noise to a basal level that was comparable to basal noise in the progenitor cells. However, each cell type exhibited a secondary minor spike at 30–35 hr, which might reflect a synchronous stimulus.

Detrended fluctuation is one method to measure expression variability, but it can suffer from distortion if the model fitting is not adequately weighted. Therefore, we also calculated the coefficient of variation, that is, the standard deviation of Yan-YFP intensity within a sliding window divided by its mean. This method yielded noise profiles with transient spikes for each cell type that was consistent with calculations using detrended fluctuation (*Figure 6—figure supplements 1 and 2*). However, while the coefficient of variation yielded results that varied strongly with bin width, the detrended fluctuations yielded profiles that were generally robust against changes in bin width (*Figure 6—figure supplement 1*).

To rule out the possibility that these unexpectedly dynamic features of Yan-YFP were caused by its transgenic origins or fusion with YFP, we compared Yan-YFP dynamics to those of endogenous Yan protein that was bound with an anti-Yan antibody. The profiles of Yan-YFP protein levels and noise were highly similar to endogenous Yan protein levels and noise, in both multipotent and differentiating cells (*Figure 6—figure supplement 3*). Thus, transient spikes of expression heterogeneity are a fundamental feature of Yan protein.

Since Yan regulates Pros expression in R7 cells, it was possible that Pros showed a transient noise spike as a consequence. Therefore, we measured Pros protein heterogeneity and found that its dynamics did not resemble that of Yan (*Figure 6—figure supplement 4*). Instead, Pros noise was

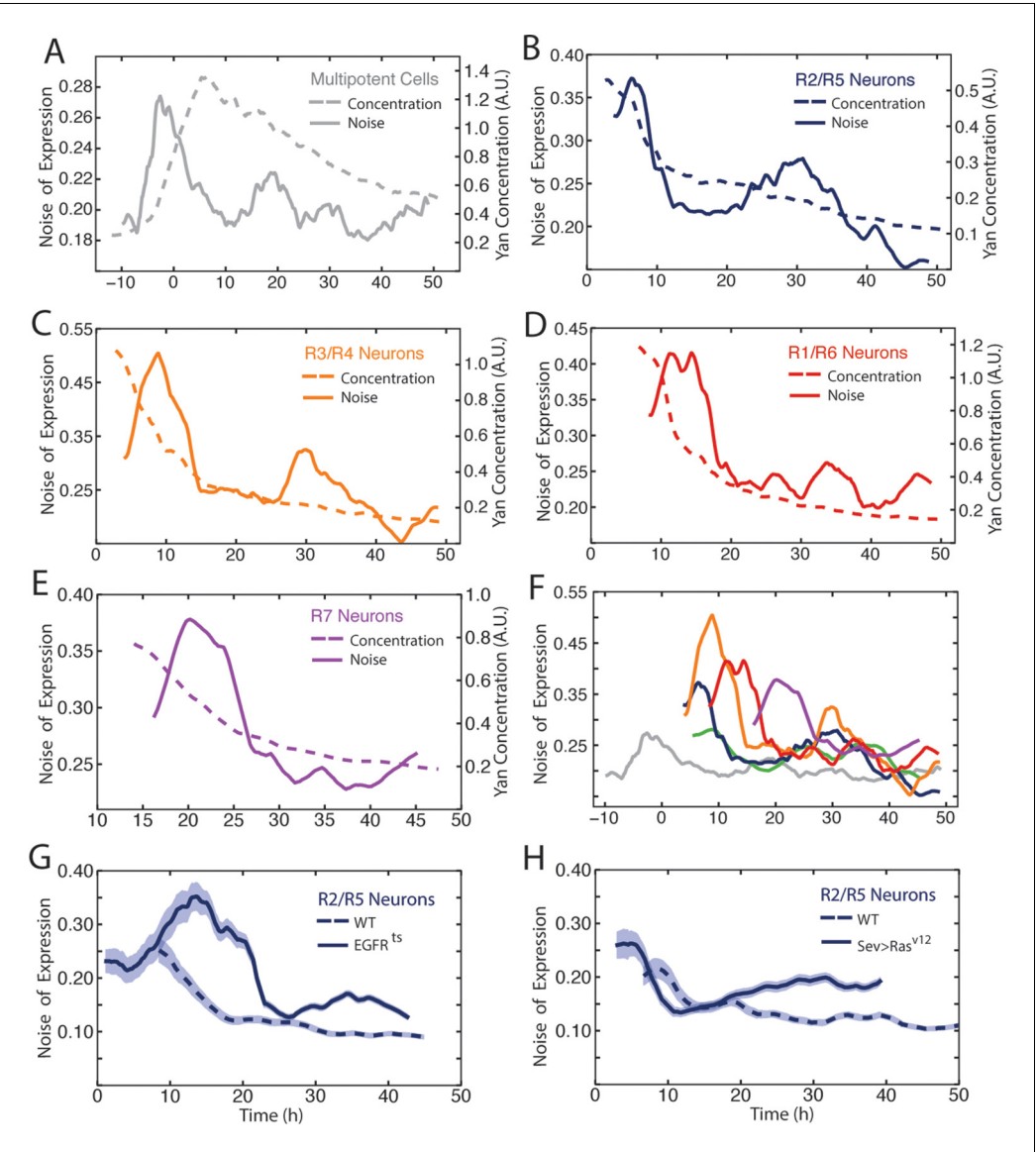

**Figure 6.** Noise in Yan-YFP expression is highly dynamic. Moving averages of Yan-YFP levels and noise (detrended fluctuations) for (**A**) progenitors, (**B**) R2/R5, (**C**) R3/R4, (**D**) R1/R6, and (**E**) R7 cells. (**F**) Comparative noise dynamics for all cells analyzed in (**A–E**). (**G**) Moving averages of Yan-YFP noise (coefficient of variation) in R2/R5 cells sampled from wildtype and *EGFR^ts* mutant eyes at the non-permissive temperature. Shown with shading is the standard error of the mean for each moving average. (**H**) Moving averages of Yan-YFP noise (coefficient of variation) in R2/R5 cells sampled from wildtype and *sev>Ras^v12* mutant eyes. Shown with shading is the standard error of the mean for each moving average.

The following figure supplements are available for figure 6:

**Figure supplement 1.** Comparison of methods to measure Yan-YFP noise based on sensitivity to window size.

**Figure supplement 2.** Measurement of Yan-YFP noise in all cell types.

**Figure supplement 3.** Comparison of Yan-YFP and endogenous Yan protein dynamics.

**Figure supplement 4.** Moving averages of anti-Pros fluorescence levels and noise (coefficient of variation) in R7 cells.

high starting at the onset of expression, and thereafter gradually declined as Pros protein levels increased in R7 cells. We conclude that noise spikes are not a general feature of gene expression in the developing eye but might reflect unique roles of Yan in mediating cell state transitions.

We wondered what might cause these spikes in Yan-YFP noise during cell state transitions. Because EGFR signaling is important for regulating Yan-YFP concentration during these transitions (*Figure 3*), we analyzed Yan-YFP noise when EGFR signaling was inhibited in *EGFR^{ts}* mutant animals raised at a non-permissive temperature. The noise spike in progenitor cells was not significantly affected by loss of EGFR signaling (data not shown). We also examined the effects of *EGFR^{ts}* on noise in differentiating photoreceptor cells. Interestingly, noise increased at the normal time of transition but the elevated noise did not quickly drop to basal levels (*Figure 6G*). Rather, high noise was extended for an additional 10 to 15 hr. Conversely, misexpressing constitutively active Ras within differentiating cells caused a premature dropdown in Yan-YFP noise (*Figure 6H*). These results indicate that EGFR/Ras signaling is required for the rapid drop in Yan-YFP noise after it has peaked, creating a transient spike.

## Discussion

This study relied upon a set of methods that enable systems-level analysis of Yan expression dynamics in a developing animal tissue. Transgene recombineering was used to insert YFP into a genomic rescue fragment of the *yan* gene, which fully replaced endogenous *yan*. Yan-YFP protein was quantified in thousands of individual cells by fluorescence confocal microscopy and automated cell segmentation analysis. Based on the unique features of *Drosophila* eye development, every analyzed cell was assigned an age, and composites of cells across a time-spectrum of ages were derived. This allowed us to reconstruct the temporal dynamics of Yan protein expression in cells as they transited from one state to another or were maintained in a given state. The fact that both Yan concentration and noise were easily measured using our approach indicates that it provides a powerful method for studying how other molecular determinants regulate cell states.

Contrary to what is currently believed, the expression of Yan in progenitor cells has many hallmarks of monostability. A stable basal state exists in cells anterior to the furrow, and when the furrow passes, Yan rises and falls to form a biphasic profile (*Figure 7A*). If cells transit towards differentiation, then the fall in Yan is accelerated but the fundamental biphasic profile is preserved. This monostable-like behavior is not like a behavior where progenitor cells exist in a high Yan stable-state and switch to a low Yan stable-state when they transit towards differentiation (*Figure 7A*). Other lines of evidence also point away from a bistable switch mechanism. Yan reaches its basal steady state many hours after cells have adopted their differentiated photoreceptor state and are executing specialized gene expression programs. Thus, Yan levels are variable at the time when cells actually become differentiated.

Since cell states are indifferent to Yan stability, it would suggest that absolute Yan levels do not dictate Yan's effects on cell behavior. This conclusion is bolstered by experiments in which *yan* gene dosage affects absolute Yan levels but not cell behavior. Several mechanisms could explain how Yan regulates cells in a concentration-independent manner. First, total Yan might vary but a specific modified form of Yan might remain constant. For example, MAPK phosphorylation of Yan could operate under Michaelis-Menten saturation to generate constant levels of phospho-Yan that depend upon MAPK activity. Second, Yan's transcriptional activities might be independent of Yan concentration due to limiting levels of other transcription factors such as Lozenge, Glass, and Sine Oculis, which are known to act combinatorially with Yan to regulate gene transcription (*Flores et al., 2000*; *Hayashi et al., 2008*; *Xu et al., 2000*). Third, cells might sense relative changes in Yan, and respond to a constant fold-change in Yan levels using integral negative feedback or incoherent feedforward loops (*Cohen-Saidon et al., 2009*; *Goentoro and Kirschner, 2009*; *Goentoro et al., 2009*; *Wartlick et al., 2011*). Indeed, Yan is predicted to function in an incoherent feedforward loop with Pnt. Pnt directly activates transcription of target genes such as *pros* and *mir-7*, but it also activates expression of Yan (*Figure 3G*), which in turn, directly represses transcription of *pros* and *mir-7* (*Li et al., 2009*; *Xu et al., 2000*).

The most intriguing hypothesis, however, is that variation in Yan concentration might actually be exploited by cells to guide their state transition. Indeed, it has recently been noted that undifferentiated stem cells express highly variable levels of transcription factors such as Nanog, Myc, Otx2, and

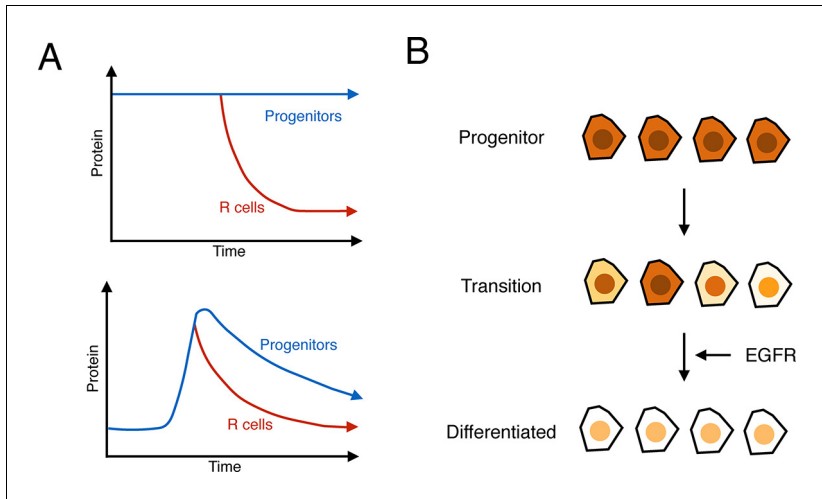

**Figure 7.** Summary of analysis. (**A**) Top: a hypothetical bistable behavior would be where Yan is in a stable high state within progenitor cells and in a stable low state within differentiated cells. Bottom: the observed behavior of Yan appears monostable, with both progenitors and differentiated cells in unstable Yan states. (**B**) Heterogeneity in Yan expression is maximal when progenitor cells enter a transition state that resolves to a more homogeneous differentiated state, dependent upon EGFR signaling. This heterogeneous transition state may be a primary mechanism for Yan's effect on cells, independent of the absolute Yan concentration within progenitors.

Pax6 (*Cahan and Daley, 2013*; *Kumar et al., 2014*; *Torres-Padilla and Chambers, 2014*). Conditions that repress variability keep these cells trapped in an undifferentiated state (*Kumar et al., 2014*). Noisy expression is thought to place cells into a transition state where a sub-population of cells at any given time express a particular combination of factors that spontaneously trigger differentiation (*Martinez Arias and Brickman, 2011*). Such a fluctuating transition state renders the probability of cell differentiation to be greater than zero, and thus provides cells with the ability to respond to extrinsic signals.

Consistent with this hypothesis, Yan levels not only vary over long time-scales but also show heterogeneity within a small time interval. This expression noise is reminiscent of the heterogeneity in Nanog expression observed in pluripotent stem cells. As progenitor cells transit to differentiated states, there is a sharp spike in Yan expression noise that coincides with the early stages of the transition. By analogy to the situation in embryonic stem cells, the spike in Yan noise could reflect the existence of a metastable intermediate state between the undifferentiated and differentiated states (*Figure 7B*).

A limitation to our interpretation is that because our data do not track individual cells over time, we cannot distinguish whether the expression heterogeneity reflects asynchronous fluctuations within individual cells or a wide range of stable microstates. Real-time single cell measurements will be required to distinguish between these possibilities.

The purpose of the elevated noise could be to prime cells to change states upon reception of a differentiation signal. Without elevated noise, differentiation or de-differentiation would be blocked. A prediction of this hypothesis is that differentiation signals would act downstream of noise elevation and might even be important for noise reduction once cells enter an irreversible differentiated state. Indeed, EGFR/Ras signaling is not required for elevation of Yan noise but rather is important in promoting the rapid reduction of noise back to its baseline level after differentiation. In the absence of signaling, the cell may remain trapped in the intermediate (undifferentiated) state.

The model raises several questions. What triggers the increase in the noise level of Yan as cells enter this transition state? Do other determinants such as Pnt similarly fluctuate? Do individual cells fluctuate over time or does this represent a population of cells that adopt stable microstates? And finally, there is the question of the noise spike in progenitor cells at the time of R8 formation. Does this represent another transition state that early progenitor cells enter before moving into a different progenitor state? One intriguing possibility is that it represents the transition from G1-arrest to

synchronized cell division that progenitor cells undergo in the second mitotic wave immediately posterior to the furrow (*Wolff and Ready, 1993*). Loss-of-function studies have suggested that Yan prevents progenitors from undergoing multiple rounds of cell division rather than again arresting at G1 phase after one round of division (*Rogge et al., 1995*). Further studies should help elucidate the causes and functions of this and the other noise peaks.

## Materials and methods

### Genetics

Growth conditions were 21°C unless stated otherwise. The recombineered *Yan-YFP* BAC transgene was previously described (*Webber et al., 2013*). It was inserted into the attP2 (3L) and attP16 (2R) sites in the genome. 2 x *Yan-YFP* animals used for most experiments were null for endogenous *yan* and carried a single copy of the *His2Av-mRFP* transgene ($yan^{ER443}/yan^{E884}$; *Yan-YFP/Yan-YFP, His2Av-mRFP*). EGFR activity was conditionally reduced by placing $Egfr^{f24}$ (*Clifford and Schupbach, 1989*) *in trans* to the ts mutant $Egfr^{tsla}$ (*Kumar et al., 1998*). Flies were raised at the permissive temperature (18°C) and shifted to a semi-restrictive temperature (26.5°C) as third instar larvae for 18-20h. At this temperature, developing eye cells had compromised EGFR activity since animals that were transferred back to the permissive temperature and allowed to eclose had rough eyes. In $EGFR^{ts}$ eye discs, there were signs of some cells undergoing apoptosis - a significant reduction of nuclear diameter, a strong Yan-YFP brightness, and anomalous nuclear position along the apical-basal axis. Apoptosis was more prevalent in discs from animals treated at temperatures greater than 26.5°C. Therefore, we chose this temperature for EGFR activity reduction so as to minimize apoptosis but still achieve effects on cell differentiation. We only included in our analysis cells corresponding to classical anatomical positions and apical basal migration patterns. Ras activation was achieved using a transgene expressing a $Ras1^{V12}$ mutant. Expression was under the control of the 3xsev enhancer and promoter (*Fortini et al., 1992*). Clones of the $pnt^2$ pan-isoform null allele were generated with *eyFLP122* and FRT82B crossover points. $Pnt^+$ tissue was labeled using *Ubi>mRFPnls*. *UAS-PntP1* and *UAS-PntP2::VP16* expression was driven with *LongGMR-Gal4* (*Wernet et al., 2003*). Dynamics of Yan using different *yan* gene copy numbers were measured in 2 x *yan* flies ($yan^{ER443}/yan^{E884}$; *Yan-YFP, His2Av-mRFP/Yan-YFP*), 4 x *yan* flies (*Yan-YFP, His2Av-mRFP/Yan-YFP*), and 6 x *yan* flies (*Yan-YFP/Yan-YFP; Yan-YFP, His2Av-mRFP/Yan-YFP*).

### Microscopy

Eye-antennal discs from white prepupae were fixed in 4% PFA/PBS for ~45 min. Discs were incubated in 1:1 (v/v) PBS:VectaShield (Vector Laboratories) for 45 min, followed by a 45 min incubation in 100% VectaShield, and then were mounted. For certain experiments, Yan and Prospero protein immunofluorochemistry was performed using mouse anti-Yan (DSHB, 1:200 dilution) and anti-Prospero (DSHB, 1:100 dilution) antibodies, and secondary goat anti-mouse Pacific Blue or goat anti-mouse Alexa 633 (Life Technologies, both at 1:200 dilution). To validate cell-type identification, discs were treated with mouse anti-Rough (DSHB), mouse anti-Cut (DSHB), rabbit anti-Senseless (Hugo Bellen), and rat anti-Elav (DSHB), and secondary Pacific Blue labeled antibodies (Life Technologies, 1:200 dilution). Samples were kept in the dark at -20°C and imaged no later than 18 hr after fixation. All discs for a given condition were fixed, mounted, and imaged in parallel to reduce measurement error.

Mounted samples were imaged with an SP5 Leica confocal microscope using a 40X objective, and the 405 nm, 514 nm, 541 nm, and 630 nm lasers were used to excite Pacific Blue, Yan-YFP, His2Av-mRFP and Alexa 633, respectively. Offset was set to zero. The gain was adjusted using the simple scanning mode without line averaging such that the brightest nuclei in the sample did not overexpose in the detector. Stacks of ~60 optical sections (1 μm) were acquired after orienting the eye disc equatorial region parallel to the x-axis of the image. Sections of 2048 x 2048 pixel resolution were collected using the 8-line average bidirectional scanning mode. A typical nucleus was usually represented by 150–300 pixels at its widest plane. Zoom and all other imaging parameters were kept constant for all samples within the same experiment. Bleaching controls were made at the beginning and end of imaging to ensure that fluorophore bleaching was not significant.

Images were acquired such that the eye's equator was parallel to the x-axis of the image, and the eye's MF was parallel to the y-axis of the image. We cropped images so that each region of interest (ROI) was a rectangle with its long axis centered at the equator (*Figure 1A*). This region extended eight ommatidial rows on either side of the equator. It was bounded on the anterior side 30–60 µm ahead of the MF and on the posterior side two ommatidial columns from the posterior margin. It is known that the margins of the eye disc exhibit different MF dynamics (*Dominguez and Hafen, 1997*; *Legent and Treisman, 2008*), and so to minimize heterogeneity due to margin effects, we limited analysis to this centered region.

## Illumination biases, autofluorescence and background correction

We estimated the potential illumination biases of the SP5 confocal microscope by exciting and imaging VectaShield media alone. We divided the images into squares of 32 × 32 pixels. For each square, we calculated and plotted the average fluorescence in 1 and 2 dimensions. Based on this analysis, we found that the microscope had no significant illumination biases across the field of view.

Previous studies of *Drosophila* embryos found that autofluorescence, which contributes to the measured total fluorescence, has to be subtracted to properly estimate the fluorescent protein signal of interest (*Surkova et al., 2008*). To measure autofluorescence of eye disc samples, we fixed and imaged wildtype discs using the same conditions as for Yan-YFP discs. The wildtype discs had negligible fluorescence. Therefore, we did not subtract autofluorescence background from total fluorescence in Yan-YFP discs.

We tested whether His2Av-mRFP fluorescence affected the measurement of Yan-YFP signal in the yellow channel. To measure the contribution of mRFP to the yellow channel, discs expressing only His2Av-mRFP were imaged with the same parameters as discs described above. We calculated the average fluorescence in 2 dimensions using a heat map to represent intensity (*Figure 1—figure supplement 4A–C*). To determine if the low RFP signal detected in the yellow channel was uniform, we segmented nuclei and measured the ratio of yellow:red signals (*Figure 1—figure supplement 4D*). We found that no more than 5% of total RFP fluorescence was detected in the yellow channel. Therefore, a complete absence of YFP fluorescence would generate 0.05 units of normalized fluorescence (yellow/red) in Yan-YFP discs.

## Analysis pipeline

A pipeline was custom-built to automatically segment nuclei and manually assign their identities with GUI software. The pipeline is accessible for download at https://dl.dropboxusercontent.com/u/2649235/Pipeline_eye_eLife.zip. Segmentation of nuclei in each optical section image was performed with a custom-built Matlab program (*Qi et al., 2013*). A detailed description of the algorithm, coding, and analysis will be published separately. We briefly describe it as follows. First, graph-cut estimates the mean overall signal and partitions the image into background and foreground, resulting in nuclear pixels clustering in the foreground. Such clusters are then analyzed with a mean-shift algorithm, which further partitions the objects into smaller pixel clusters in a locally optimized manner. Due to the tight packing of eye disc nuclei, many such segmented objects contain multiple nuclei whose fluorescence signals partially overlap. Thus, objects are then subjected to concavity-convexity analysis. According to this analysis a cluster of cells is split into multiple cells, based on a threshold that quantifies the degree of concavity of the cluster (*Figure 1—figure supplement 2*). The Matlab implementation of this algorithm contains 5 parameters.

| Parameter | Range | Default value | Parameter description and function |
|---|---|---|---|
| Intensity range | 1–20 | P1=7 | Parameter used to define the range of similar pixels based on their intensity values (the higher the parameter value, the smaller the number of objects found). Works in combination with parameter 3 (spatial range). |
| Window size | 5–250 | P2=30 | Parameter has to be smaller than the diameter of an individual cell. Objects smaller than this size will be merged with its nearest neighbor. |

| | | | | |
|---|---|---|---|---|
| Spatial range | 0–20 | P3=10 | | Parameter used to define the range of similar pixels based on their geometry. The smaller the value of the parameter the fewer the pixels which will become part of the segmented nuclei. Works in combination with parameter 1 (intensity range). |
| Concave- convexity threshold | 1–20 | P4=5 | | Parameter used to quantify the concavity of an object. Based on its value a decision is made as to whether to split an object into multiple ones. The larger the value, the less likely a cut will be made. |
| Noise intensity | 1–140 | P5=30 | | Parameter quantifying our prior-knowledge on the noise intensity level in the image. The larger the value of this parameter the higher (brighter) the background noise. |

The developed algorithm was evaluated extensively utilizing publicly available hand-segmented benchmark datasets of microscope images. Towards this task a number of objective performance metrics were utilized, such as the Rand Index and the Hausdorff distance. In facilitating the evaluation of the various indices we utilized the following error classes: 1) two reference nuclei ("ground truth") are assigned to a single machine-segmented nucleus; 2) one reference nucleus is segmented as two by the machine; 3) machine picks up a non-existent nucleus from the background; 4) a nucleus belonging to the reference image is lost in the machine-segmented image. In all experiments the developed algorithm outperformed existing segmentation algorithms that are widely used and referenced, such as the watershed algorithm. For example, with respect to the Rand index, the performance of the developed algorithm was 91% as compared to 78% achieved by the watershed algorithm (data not shown). The developed algorithm was also compared against manual segmentations of the same eye disk images. In all cases it resulted in segmentations very similar to the manual ones.

While performing manual segmentation, we observed that the mean and standard deviation of pixel intensity for a nucleus (size >80 pixels) was distorted by pixels close to the segmented boundary. This is due to light scattering. The phenomenon has also been reported for *Drosophila* embryonic nuclei *(Surkova et al., 2008).* Thus, once the automated segmentation routine was completed, the contour of each object was shrunk using two shrink parameters - shrink and shrink level. The first parameter determines whether boundary shrinking occurs, and the second parameter determines the number of pixels from the boundary to be eroded. We find that this operation results in smaller measurement errors. A single set of parameters (P1=6, P2=30, P3=3, P4=5, P5=30, Shrink=1, Shrinklevel=4) was used for all layers. This parameter combination produced accurate segmentations, despite the heterogeneity in size and shape of different cell types.

We assigned cell-type identities to segmented nuclei by using nuclear position and morphology, two key features that enable one to unambiguously identify eye cell types without the need for cell-specific markers (*Wolff and Ready, 1993*). To perform this task, we displayed the confocal microscopy data in a custom-made Graphic User Interface (GUI), which allowed users to visualize the contours of segmented nuclei mapped on section images, and for users to label each segmented nucleus with an ID value. Each ID value was then automatically assigned to the record of fluorescence intensities and positional information for the segmented object within the database (*Figure 1—figure supplement 2*). Our facilitated method of manual identification was over 98% accurate in identifying cell types (*Figure 1—figure supplement 3*). We plotted the centroid positions of identified cell types on a 2D Cartesian plane. These produced coordinate maps that are in complete accordance with anatomical descriptions of the eye disc (*Figure 2—figure supplement 1*).

For each cell that we identified, we only used the data obtained from the optical section with the greatest nuclear contour. Consecutive sections in a stack could easily be scrolled through to ascertain the section having the greatest 2D nuclear contour for any given cell. This approach prevented oversampling of cells and minimized measurement error due to insufficient pixel number in a nuclear section. We tested the veracity of this 2D sampling method by comparing Yan-YFP fluorescence

intensity in different optical sections of the same nucleus. If fluorescence is uniformly distributed throughout the nucleus volume, then fluorescence intensities will be similar across different optical sections. This was indeed found to be the case when we performed such sampling on 20 distinct nuclei belonging to various cell types.

## Fluorescence normalization

We found that His2Av-mRFP exhibits non-uniform fluorescence intensity along the anterior-posterior axis of the eye disc (*Figure 1—figure supplement 4E*). Intensity particularly fluctuates in regions immediately posterior to the MF. We also observed fluctuations in nuclear size along the same axis (*Figure 1—figure supplement 4F*). These fluctuations are somewhat anti-correlated with His2Av-mRFP intensity fluctuations. It suggested that perhaps nuclear volume dynamics are responsible for changes in His2Av-mRFP concentration and therefore fluorescence intensity. Hence, we multiplied fluorescence intensity by nuclear size to provide a measure of His2Av-mRPF content (mass), which we predicted should be constant in G1-arrested cells. Indeed, His2Av-mRPF content was more uniform in cells along the anterior-posterior axis (*Figure 1—figure supplement 4G*). However, there were two regions in which His2Av-mRFP content was greater: one region anterior to the MF and another region immediately posterior to the MF. These correspond to regions in which many cells are proliferative and therefore can be in S and G2 phases. The anterior region is where asynchronous division occurs and the posterior region is where the second mitotic wave occurs (*Wolff and Ready, 1993*).

We reasoned that some of the variation in Yan-YFP intensity that we observe could be attributed to the same factors that cause His2Av-mRFP variation —DNA ploidy and nuclear volume. Since we wanted to measure Yan-YFP output as a function of regulatory network activity and not DNA content or volume, we normalized the fluorescence intensity of Yan-YFP to the fluorescence intensity of His2Av-mRFP. Normalization was done by taking the ratio of the mean pixel intensity of YFP over mean pixel intensity of RFP for each segmented nucleus.

## Conversion of cell position to developmental time

For each disc, we calculated a conversion factor that makes equivalent the distance travelled by the MF to the time required by the MF to travel that distance. This conversion is possible for several reasons. First, the MF moves at constant velocity, making a new column of R8 neurons every 2 hr at 21°C (*Campos-Ortega and Hofbauer, 1977*). Second, no cell migration occurs. There is a region where cell division occurs posterior to the MF —the second mitotic wave. Although this could potentially distort the physical distance between ommatidial columns, we find that it does not create a significant displacement. As a result, distances between R8 cells in adjacent columns do not undergo major rearrangements.

The conversion factor was derived by finding the average distance between adjacent columns and relating this distance to the two hour time interval required to form a new column. Using a Delaunay triangulation, we determined the network of R8 cells within each sample (*Figure 2—figure supplement 1I*). In this network, nodes are R8 nuclei centroids and links are the first R8 neighbors of each R8 cell nucleus found in the triangulation. The Delauney triangulation for a set P of points in a plane is a triangulation DT(P) in which no point in P is inside the circumcircle of any triangle in DT(P), and in which the minimum angle of all the angles of the triangles in the triangulation is maximized. As a result, Delauney triangulations tend to avoid skinny triangles, thus providing a proper estimation of the grid of nearest neighbors for each R8 neuron in the set. We enumerated all the R8 neighbors for each R8 cell found in the triangulation, and for each R8 cell, we selected neighbors whose links to the R8 cell were oriented between 30° to 60° away from the anterior-posterior axis (*Figure 2—figure supplement 1J*). We reasoned that these neighbors would correspond to R8 cells found within adjacent columns. The pairing process was repeated for every R8 cell in the network. The anterior-most column of R8 cells did not pair with more anterior cells since there were none. For each R8 pair, we decomposed the diagonal distance separating the two R8 cells using *Pythagoras*. We used the x-component of the distance separating the two R8 cells as the distance traveled by the furrow between those two adjacent columns. We then computed the average distance ($\mu$) in pixels that the MF travels between adjacent columns as

$$\mu = \frac{1}{n}\sum_{i=1}^{n}|x_{ia} - x_{ib}|$$

where $x_{ia}$ and $x_{ib}$ are the $x$ coordinates in pixel values for the $i^{th}$ pair of R8 cell centroids, and $n$ is the total number of R8 pairs. The distance $\mu$ and the two hours required for the MF to travel it, then allowed us to convert the distance that any disc cell was from the first column into a developmental time point for that cell:

$$t = \frac{2}{\mu}(x_C - x_{C1})$$

where $t$ is the developmental time of cell $c$ (in hours), $x_c$ is the $x$ coordinate of the cell $c$ centroid (in pixels), and $x_{C1}$ is the $x$ coordinate for the first column (in pixels).

## Sample alignment and analysis

We analyzed a minimum of four replicate eye discs for each treatment. A moving line average was generated for Yan-YFP intensity in progenitor cells for each disc sample of a given treatment. Samples from the same treatment were then aligned along the time axis such that the line average inflection points of Yan-YFP in the MF were minimized between all samples. We found that alignment using only the Yan-YFP inflection point caused the first columns of all samples to align with one another. It also caused the first appearance of photoreceptors in all samples to align with one another. Although we present the results of pooled samples and their analyses, these findings do not depend on sample alignment, and are reproducibly observable in individual samples when analyzed separately. A typical sample contained >1500 measured and cell-type assigned nuclei. Thus, each treatment represents close to 6000 datapoints.

After samples were pooled according to treatment, we calculated a moving line average for each cell type class using a moving window of fixed size. We used different window sizes for progenitors (n=130 cells) and differentiating cells (n=40), and their output was smoothened with a second-order sliding window (n=20).

We partitioned the trajectory of Yan-YFP in multipotent progenitors into two independent phases: induction and decay. We used a Hill function to fit the induction phase:

$$Y(t) = \frac{a + bt^n}{t^n + k^n}$$

where $Y(t)$ is Yan-YFP intensity at time $t$, $a$ is the intensity at $t=0$, $b$ is a constant, $n$ is the Hill coefficient, and $k$ is the time when $Y$ is half-maximal. We implemented the *minimize* routine in scy.py to minimize the least squares between the data and a model estimating the values of $a$, $b$, $k$ and $n$ to find the best fit to the data (*Figure 4—figure supplement 2A*).

To fit the decay phase of Yan-YFP, we tested linear, exponential, and polynomial functions, and we found that the simplest function that best fit all of the data was an exponential function (*Figure 4—figure supplement 1*). The exponential decay function used is:

$$Y(t) = A + Be^{-(t/\tau)}$$

where $Y(t)$ is Yan-YFP intensity at time $t$, $A$ is the intensity at $t=\infty$, $B$ is an amplification constant, and $\tau$ is the mean lifetime. These parameters were estimated using the *minimize* routine (*Figure 4—figure supplement 2A–F*). To obtain interval estimates for these parameters, we implemented a bootstrapping routine in which 1000 fits were iteratively performed with 70% of the data randomly selected for each fit, performing data resampling without replacement. No significant difference was found in the results when resampling was done with replacement. For each fit, we calculated the half-life as

$$T_{\frac{1}{2}} = \tau ln2$$

The mean half-life and uncertainty estimates were calculated for bootstrapped samples (*Figure 5—figure supplement 1*).

Yan-YFP noise was calculated one of two independent ways. First, we computed the coefficient of variation (CV = sigma/mean) inside a sliding window. The window size or bin width was varied for

different cell types to determine if the CV profile significantly changed (*Figure 6—figure supplement 1*). A window size of n=280 nuclei for progenitors and n=70 nuclei for differentiating cells was chosen for subsequent analysis (*Figure 6—figure supplement 2*). We calculated Pros noise in the same way. The second way we calculated Yan-YFP noise was to compare each datapoint to the fitted Hill and exponential functions, and estimate their residuals. The standard deviation of the residuals within a sliding window was divided by the mean value of the fitted function within the window, which gave the detrended fluctuation (*Goldberger et al., 2002*). The window size was varied from n=20 to n=500 nuclei, and plotted as shown (*Figure 6—figure supplement 1*). A window size of n=280 nuclei for progenitors and n=70 nuclei for differentiating cells was chosen for subsequent analysis (*Figure 6—figure supplement 2*).

### Pooling pairs of differentiated cells

Initially we carried out analysis of each cell type separately, differentiating between cell type pairs that commit almost concurrently (R2 and R5, R3 and R4, R1 and R6, C1 and C2, C3 and C4). We did not find significant differences between individuals with each pair in terms of their Yan-YFP dynamics. Therefore, we pooled the pairs to gain power in the estimations of the mean trajectory and moving averages. An exception was observed with the misexpression of PntP1, where a difference between the R3/R4 cells was observed. But since both cells behaved by changing Yan-YFP levels in the same direction, we pooled them for the analysis shown in *Figure 3*.

### Spatial correlation analysis

To quantify the local influence of R8 cells on adjacent differentiating neurons, we calculated sliding window correlations along the anterior-posterior axis (X-axis) between the Yan-YFP levels in R2 and R5 nuclei and the their physical distance to the nearest R8 nuclei. We first detrended R2/R5 Yan-YFP levels according to their positions in X to avoid spurious spatial correlations with developmental time. Specifically, we performed standard LOESS detrending in X on log-transformed R2/R5 Yan-YFP levels using local second-degree polynomials and tri-cubic weighted neighborhoods with a smoothing parameter of 0.5. The resulting residual log-transformed R2/R5 Yan-YFP levels were distributed with approximately constant variance. We separately calculated the physical distance from the center of each R2 and R5 nucleus to the center of its nearest R8 nucleus. Detrending and distance calculations were performed independently for each measured eye disc (4 WT and 4 *EGFR*[ts] replicates) and then pooled for correlation calculations. For a series of windows along the X-axis that each span 12.5 hr of developmental time, we selected R2 or R5 cells within those windows and calculated the Pearson product-moment correlation coefficient from their residual log-transformed Yan-YFP levels and distances to the nearest R8. The p-value associated with each window is calculated from the corresponding t-test assuming an uncorrelated bivariate normal distribution for sample points. The explained variance reported for each window is derived from the $R^2$ value of a standard linear fit. Correlation analysis was performed using the R statistical software package (v3.1.2).

## Acknowledgements

We are grateful to Hugo Bellen, to Christian Klambt, to Kevin White and the Recombineering Core (U. Chicago), the Northwestern Biological Imaging Facility, the Developmental Studies Hybridoma Bank (DSHB), and the Bloomington Drosophila Stock Center. We thank members of the Amaral, Carthew, Dinner and Rebay labs for helpful discussions. Special thanks to Jean-François Boisclair Lachance for advice and feedback at every stage of this project. NP was funded by the Chicago Biomedical Consortium, the Robert Lurie Comprehensive Cancer Center, and an International Student Research Fellowship from the Howard Hughes Medical Institute. This work was supported by the Department of Energy grant DE-NA0002520 (AKK), and the National Institutes of Health grants P50GM081892 (IR, AD, LANA, RWC), R01GM80372 (IR), and R01GM077581 (RWC).

## Additional information

### Funding

| Funder | Grant reference number | Author |
|---|---|---|
| National Institute of General Medical Sciences | R01GM077581 | Nicolás Peláez<br>Richard W. Carthew |
| Howard Hughes Medical Institute | Early Investigator | Nicolás Peláez<br>Arnau Gavalda-Miralles<br>Heliodoro Tejedor Navarro<br>Luís A.N. Amaral |
| U.S. Department of Energy | DE-NA0002520 | Bao Wang<br>Aggelos K. Katsaggelos |
| National Institute of General Medical Sciences | R01GM80372 | Ilaria Rebay |
| National Institute of General Medical Sciences | P50GM081892 | Nicolás Peláez<br>Ilaria Rebay<br>Aaron R. Dinner<br>Richard W. Carthew |

The funders had no role in study design, data collection and interpretation, or the decision to submit the work for publication.

### Author contributions

NP, Conception and design; Acquisition of data; Analysis and interpretation of data; Drafting or revising the article; AGM, Analysis and interpretation of data; Drafting or revising the article; BW, HTN, AKK, Acquisition of data, Drafting or revising the article; HG, ARD, Analysis and interpretation of data, Drafting or revising the article; IR, RWC, Conception and design, Analysis and interpretation of data, Drafting or revising the article; LANA, Conception and design; Analysis and interpretation of data; Drafting or revising the article

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
