## [Decision Letter]

Thank you for submitting your work entitled "Dynamics and Heterogeneity of a Fate Determinant During Transition Towards Cell Differentiation" for peer review at *eLife*. Your submission has been favorably evaluated by Naama Barkai (Senior editor) and three reviewers.

The reviewers have discussed the reviews with one another and the Reviewing editor has drafted this decision to help you prepare a revised submission.

Summary:

The paper analyzes the expression of Yan, a regulator of photo-receptor differentiation in the *Drosophila* eye imaginal disc. The major claim is that Yan controls differentiation not through its absolute levels, but through its noisy expression. To get to this conclusion, the authors show that (1) Yan levels are not bi-modal (as was claimed before) but form a wide, continuous distribution, (2) Yan's level show highest fluctuations at the time of cell differentiation, and subsequently decrease, (3) increasing Yan concentrations does not affect fate decisions, and (5) genetic manipulations that reduce noise (among other effects) influence fate decisions.

All reviewers were excited by the notion that stochastic levels of Yan may play an important mechanism in eye differentiation. However, it was also agreed that several controls are absolutely essential to make sure that the data supports these claims.

Essential revisions:

1) Compare Yan-YFP to endogenous Yan. This is particularly important to validate the gradual (vs. the previously reported threshold-like) expression of Yan.

2) Check the accuracy by which you identify the correct cell-types.

3) Compare the noisiness of Yan expression to other key regulators (e.g. Pros).

Please see detailed reviews below for more details about the controls above. Also, please try to address the remaining points, at least refer to them in your writing.

*Reviewer #1:* The authors have generated a fully functional YFP-tagged version of the ETS domain transcription factor Yan and studied Yan protein dynamics in the *Drosophila* eye disc during differentiation of photoreceptor and cone cells. Although the authors cannot perform live imaging with eye discs they use the stereotypic sequence of photoreceptor and cone cell differentiation to gain temporal information on how Yan expression changes during the transition from multipotency to the differentiated state. In variance to earlier reports the authors do not find evidence for a bistable switch with high levels of Yan expression in progenitors and low levels in differentiated cells. They rather observe a monostable concentration curve with interesting stochastic properties. The curve shows highest fluctuations in concentration when the cells begin to differentiate. Fluctuation decrease when the differentiated state is reached. Changing overall Yan concentrations (by increasing the copy number of the construct) does not affect the dynamics and noisiness and has no impact on fate decisions. Thus, Yan does not act as a concentration-dependent (morphogen-type) transcription factor. The authors suggest that rather the noisiness itself might play a functional role: high levels of noise might be functionally relevant for the transition from multi- to restricted potency. Genetic manipulations are presented which support this view. Reduction of EGF signaling delays the reduction of noise during differentiation and constitutive EGF activation (activated Ras) reduces noise even in progenitors. Both manipulations influence fate decisions.

Altogether this is a very impressive study both with regard to technical sophistication and to conceptual design. The paper is very well written.

Nonetheless, general concerns remain:

1) An important limitation of the paper results from the fact that no techniques are available for following Yan expression in single cells over time. Thus, the nature of the fluctuations remains not resolved. The authors are aware of this weakness.

2) The central hypothesis of the paper (Yan fluctuations are functionally relevant) can be rigorously tested only if the noisiness of Yan expression is manipulated independently from EGF signaling, e.g. by manipulating transcriptional of translational control elements of Yan. To clarify both points would require a considerable amount of additional experimentation.

Even in its current version, however, the paper definitely enters uncharted territory and provides evidence for a novel mechanism, which might be of general relevance. Transcription factor stochasticity has recently been observed in other systems (e.g. mammalian embryonic stem cells). The paper shows that the *Drosophila* eye disc represents an experimental system to study this new and exciting phenomenon, the mechanism of which is totally unknown. Therefore, I vote for publication in *eLife*.

*Reviewer #2:* In this review, the authors conduct an analysis of expression of the Yan transcription factor during differentiation of *Drosophila* retinal cells. They found that Yan expression is noisy at the time of differentiation. The temporal dynamics of this noise are susceptible to regulation by EGFR/ras signaling. Further, Yan-mediated regulation of differentiation is concentration independent. The authors propose that their data suggests that the heterogeneity is important for the transition and that noise reduction stabilizes cell state.

This paper brings the well-studied fly eye into the age of quantitative systems biology. The finding that Yan is a "noisy" gene is interesting but it does not actually prove the authors' ultimate conclusion – that this noise is biologically important or unique. At this point, the coupling of noisy Yan and differentiation is correlative.

The authors set up the paper well in stating that they are taking single timepoint data and not live imaging. However, later, they make conclusions as if the observed heterogeneity is occurring in single cells. Since the observations are made at single timepoints, the path of Yan expression may be very reproducible and the variation may come in relative levels. The authors should clearly state how their conclusions are impacted by the single timepoint data.

From a technical aspect, the cell identification would be an impressive feat. However, the authors must perform control experiments to measure the accuracy using cell specific antibodies (Pros, Sens, etc.) or reporters. The authors did a measurement of accuracy versus their personal identification. It appears the algorithm performs at 91% (it is not totally clear if this is accuracy or a different measure). If this is the case, then 9% of the time, the program gets the cell wrong. How much of the variation is due to wrong cell ID? Highly accurate cell identification seems extremely critical. The observed noise could simply be due to getting the cell ID wrong. The authors must demonstrate that the software is accurate enough to provide meaningful information about the noisiness of gene expression.

Ultimately, this paper is moving the study of the fly eye in an exciting direction. However, the authors must provide evidence that the noisiness is biologically meaningful. Ideally, the authors would do this by dampening or increasing noise and showing a biological phenotype. Though this would be extremely exciting, it would also be challenging. Alternatively, the authors could show the uniqueness of noisy Yan expression by comparing its expression to other critical transcription factors in retinal development either in the upstream part of the network (ex: Pnt) or downstream (ex: Pros, Sens). Such an analysis, using their system, could determine whether noisiness is common in the retinal GRN or unique to Yan, which would be an interesting observation.

Additional comments:

1) The authors claim the early phase of Yan YFP decay is inhibited in R3/4 cells but this does not appear to occur in Figure 3, where WT and *EGFR^ts^* practically overlap. The authors should change their conclusion or provide statistical evidence that they are different.

2) The writing is very dense. For example, "we observed that the early phase of YAN-YFP decay was inhibited by *EGFR^ts^*…". This triple negative (decay/inhibited/ *EGFR^ts^*) is extremely challenging for the reader to interpret. The authors should work to simplify the text in general.

*Reviewer #3:* This paper makes use of a *Bac* transgene containing an in-frame YFP fusion with the Yan protein to study the dynamics of Yan protein concentrations during the process of eye differentiation in *Drosophila*. This gene fusion completely rescues a *yan* mutant, suggesting that the transgene provides all functions of the endogenous gene. Expression levels driven by the *Yan-YFP* transgene in individual cells are monitored with respect to the advancing differentiation furrow, taking advantage of the fact that cells of different stages of differentiation are spatially arranged during this process.

The major conclusions are: 1. Absolute thresholds of Yan protein are not required for normal development; 2. The levels of Yan are monophasic, not biphasic, as suggested previously; 3. There is unanticipated noise in the regions of highest expression levels of Yan; 4. This noise is critical for the proper differentiation of the major photoreceptor and cone cell fates in the developing eye; and 5. This noise is controlled in part by the EGF signaling pathway.

I think the methods for measuring Yan-YFP dynamics and the data analysis are sound, and the experiments with multiple copies of the *Yan-YFP* transgene support the first conclusion.

With respect to the other conclusions, I am very concerned about the assumption that the detailed measurements of the *Yan-YFP* transgene reflect the expression dynamics of the endogenous protein. The argument that the transgene rescues the *yan* mutant is actually quite weak, because it is possible that developing animals that can tolerate very different expression levels can also tolerate very different decay dynamics. In Figure 1, a direct comparison between Yan-YFP and staining with an anti-Yan antibody is shown. By my eye, these patterns look quite different, with one being much more stochastically variable than the other. Unfortunately, the labels on the figure and the legend do not match, so it is not clear which is which, but my guess is that the more blotchy signal is from the *YFP* transgene.

If the signals monitored from the *YFP* transgene are quite different from the endogenous protein signals, then the noisy dynamics at the heart of the paper's conclusions could be simply artifacts of stability differences between the fusion protein and the endogenous protein. It is clear that the authors have looked at anti-body staining samples, and I'm wondering whether they see the bi-phasic distributions proposed by previous groups. If so, this would challenge the major conclusions of the paper.

---

## [Author Response]

*Essential revisions:*

*1) Compare Yan-YFP to endogenous Yan. This is particularly important to validate the gradual (vs. the previously reported threshold-like) expression of Yan.*

We have compared Yan-YFP to endogenous Yan protein detected via immunofluorescence. The dynamics and heterogeneity of both proteins are highly similar to one another. A thorough quantitative comparison of the two proteins is provided in Figure 6—figure supplement 3. Based on these results, we are confident that the YFP fusion protein is a faithful reporter of the endogenous protein’s behavior.

*2) Check the accuracy by which you identify the correct cell-types.*

It appears that the reviewers thought that cell-type classification was computationally automated. It was not. All identification and classification was performed by individuals using the input confocal stack data displayed in our custom-built GUI. The app enables a user to simultaneously visualize the fluorescence channels and segmentation boundaries in each slice, and scroll between slices. Based on nuclear position and shape, we assign identities to cells.

To validate the accuracy of our identification, we used a variety of antibodies recognizing marker proteins expressed in subsets of cell types. We used a Pacific Blue fluor to visualize these marker proteins within Yan-YFP H2A-mRFP eye discs. We blindly classified cells with the Pacific Blue channel turned off, and then compared our identifications with the marker protein patterns. For all cell types, we were accurate at least 98% of the time. This data is shown in Figure 1—figure supplement 3.

*3) Compare the noisiness of Yan expression to other key regulators (e.g. Pros).*

We used anti-Pros to visualize Pros protein dynamics and noise. The results are shown in Figure 6—figure supplement 4. Strikingly, Pros expression begins with high noise and then gradually drops as Pros expression levels increase in R7 cells. This noise profile is in contrast to Yan’s noise profile, in which noise spikes from low to high to low as R7 cells differentiate. Thus, the Yan noise spike is not a general property of protein expression dynamics during eye cell differentiation.

*Please see detailed reviews below for more details about the controls above. Also, please try to address the remaining points, at least refer to them in your writing.*

Reviewer #1:

[…] Nonetheless, general concerns remain:

1) An important limitation of the paper results from the fact that no techniques are available for following Yan expression in single cells over time. Thus, the nature of the fluctuations remains not resolved. The authors are aware of this weakness.

We had noted this limitation in the Results and now expand upon this limitation in the Discussion.

*2) The central hypothesis of the paper (Yan fluctuations are functionally relevant) can be rigorously tested only if the noisiness of Yan expression is manipulated independently from EGF signaling, e.g. by manipulating transcriptional of translational control elements of Yan. To clarify both points would require a considerable amount of additional experimentation.*

We agree with the reviewer and plan to experimentally test the hypothesis in ways similar to the ones described.

Reviewer #2:

*[…] The authors set up the paper well in stating that they are taking single timepoint data and not live imaging. However, later, they make conclusions as if the observed heterogeneity is occurring in single cells. Since the observations are made at single timepoints, the path of Yan expression may be very reproducible and the variation may come in relative levels. The authors should clearly state how their conclusions are impacted by the single timepoint data.*

We have now stated in the Discussion that the heterogeneity might either be the result of fluctuations in single cells over time or divergence in the trajectories of Yan protein dynamics between cells. Our observations cannot distinguish between these two possibilities.

*From a technical aspect, the cell identification would be an impressive feat. However, the authors must perform control experiments to measure the accuracy using cell specific antibodies (Pros, Sens, etc.) or reporters. The authors did a measurement of accuracy versus their personal identification. It appears the algorithm performs at 91% (it is not totally clear if this is accuracy or a different measure). If this is the case, then 9% of the time, the program gets the cell wrong. How much of the variation is due to wrong cell ID? Highly accurate cell identification seems extremely critical. The observed noise could simply be due to getting the cell ID wrong. The authors must demonstrate that the software is accurate enough to provide meaningful information about the noisiness of gene expression.*

As stated above, we have done this.

*Ultimately, this paper is moving the study of the fly eye in an exciting direction. However, the authors must provide evidence that the noisiness is biologically meaningful. Ideally, the authors would do this by dampening or increasing noise and showing a biological phenotype. Though this would be extremely exciting, it would also be challenging. Alternatively, the authors could show the uniqueness of noisy Yan expression by comparing its expression to other critical transcription factors in retinal development either in the upstream part of the network (ex: Pnt) or downstream (ex: Pros, Sens). Such an analysis, using their system, could determine whether noisiness is common in the retinal GRN or unique to Yan, which would be an interesting observation.*

As stated above, we did this for Pros, and the noise profile of Pros does not resemble Yan’s profile.

*Additional comments:*

*1) The authors claim the early phase of Yan YFP decay is inhibited in R3/4 cells but this does not appear to occur in Figure 3, where WT and* EGFR^ts^*practically overlap. The authors should change their conclusion or provide statistical evidence that they are different.*

We have deleted Figure 3 which showed analysis of R3/4 cells. Uncertainty estimates are now provided for the analysis of multipotent and R2/5 cells in Figure 3.

*2) The writing is very dense. For example, "we observed that the early phase of YAN-YFP decay was inhibited by EGFR^ts^…". This triple negative (decay/inhibited/ EGFR^ts^) is extremely challenging for the reader to interpret. The authors should work to simplify the text in general.*

We have changed the awkward sentences as suggested.

Reviewer #3:

[*…] With respect to the other conclusions, I am very concerned about the assumption that the detailed measurements of the* Yan-YFP *transgene reflect the expression dynamics of the endogenous protein. The argument that the transgene rescues the* yan *mutant is actually quite weak, because it is possible that developing animals that can tolerate very different expression levels can also tolerate very different decay dynamics. In Figure 1, a direct comparison between Yan-YFP and staining with an anti-Yan antibody is shown. By my eye, these patterns look quite different, with one being much more stochastically variable than the other. Unfortunately, the labels on the figure and the legend do not match, so it is not clear which is which, but my guess is that the more blotchy signal is from the* YFP *transgene.*

We have corrected the mislabeling in Figure 1. As stated above, we have quantitatively compared Yan-YFP to endogenous Yan.